# The Fibroblast Growth Factor Receptors in Breast Cancer: from Oncogenesis to Better Treatments

**DOI:** 10.3390/ijms21062011

**Published:** 2020-03-16

**Authors:** Navid Sobhani, Chunmei Fan, Pedro O. Flores-Villanueva, Daniele Generali, Yong Li

**Affiliations:** 1Section of Epidemiology and Population Science, Department of Medicine, Baylor College of Medicine, 1 Baylor Plaza, Houston, TX 77030, USA; Chunmei.Fan@bcm.edu (C.F.); PedroOmar.Flores-Villanueva@bcm.edu (P.O.F.-V.); 2Department of Medical, Surgical and Health Sciences, University of Trieste, Cattinara Hospital, Strada Di Fiume 447, 34149 Trieste, Italy; dgenerali@units.it

**Keywords:** metastatic breast cancer, targeted therapies, fibroblast growth factors’ receptors drugs

## Abstract

Breast cancer (BC) is the most frequent form of malignancy and second only to lung cancer as cause of deaths in women. Notwithstanding many progresses made in the field, metastatic BC has a very poor prognosis. As therapies are becoming more personalized to meet the needs of patients, a better knowledge of the molecular biology leading to the disease unfolds the possibility to project more precise compounds or antibodies targeting definite alteration at the molecular level and functioning on such cancer-causing molecules expressed in cancer cells of patients, or present as antigens on the surface of cancer cell membranes. Fibroblast growth factor receptor (FGFR) is one of such druggable targets, activated by its own ligands -namely the Fibroblast Growth Factors (FGFs). This pathway provides a vast range of interesting molecular targets pursued at different levels of clinical investigation. Herein we provide an update on the knowledge of genetic alterations of the receptors in breast cancer, their role in tumorigenesis and the most recent drugs against this particular receptor for the treatment of the disease.

## 1. The Receptor

### 1.1. Biochemistry of the Receptor

As to the epidemiology of the disease, Breast Cancer (BC) is a devastating cancer in women. Worldwide it is first in the list in terms of frequency of all tumors and it is second only to lung cancer in terms of mortality. In the year 2019, it has been recorded a total of 271,270 newly diagnosed breast cancer patients—of whom 41,488 died—in the United States [1]. As personalized medicine advances, stratification of patients based, on molecular targets, is becoming the standard of care. FGFRs harbor cancer-driver mutations that can be targeted with antibodies for the treatment of BC.

Briefly, FGFRs are receptor tyrosine kinase members, encompassing the cellular membrane in a single region and therefore considered single-pass membrane proteins [2]. They are made of N-terminal extracellular domains with three different immunoglobulin-like subdomains (D1, D2 and D3), a transmembrane domain constituted of an α-helix, and an intracellular region, which has tyrosine kinase motifs capable of phosphorylation and a carboxyl-terminal end. The FGFR family, in humans, consists of six receptors (FGFR1-6), bound by 18 ligands called fibroblast growth factors (FGFs). The *FGFR1* gene is located on chromosome 18p11.23, while *FGFR2* is on chromosome 10q26.13, *FGFR3* is on chromosome 4p16.3, *FGFR4* is on chromosome 5q35.2, *FGFR5* is on chromosome 4p16.3, and finally *FGFR6* is located on chromosome 6p21.33 (also called Fibroblast Growth Factor like-1). *FGFR* 1, 2 and 3 alternative splicing may encode alternative isoforms with different ligand binding specificities [3]. In addition to its involvement in cancer, like many other oncogenic drivers, the receptor is also important for the development of the skeletal system [4,5], the metabolism and embryogenesis [6,7]. Interestingly, Fibroblast Growth Factors (FGFs) engage with many co-factor, such as heparin or heparan sulfate proteoglycan (HSPGs) at the cell surface, increasing the affinity binding to cellular FGFR [8,9,10]. Indeed, FGFs, after being secreted are almost immediately taken up by HSPGs. HSPGs, in turn, stabilize the interaction between FGF ligand and FGFR [11] by safeguarding FGFs from degradation by proteases [12]. Ligands have different specificity in binding to FGFRs; some of them bind to different receptors, such as FGF1, while others, such as FGF6, bind only to one receptor isoform [13].

### 1.2. FGFR Signalling

RTKs were discovered about fifty years ago. Since then, their signal transduction has been explained through the canonical or diffusion model [14]. It is known that cell membrane receptors are responding to a signal that is been transmitted from outside the cell through molecules that bind to them. Once bound the receptor becomes activated and it triggers a downstream series of events that activate other molecules. RTKs are the largest class of such receptors showing such capability. Briefly, ligand-binding causes RTKs monomers to form dimers and this tethers tyrosine residues of the monomers close to each other, which thereby cross-phosphorylate and, as a direct consequence, activating each other [2,15]. It is through this mechanism of dimerization and cross-phosphorylation that other molecules nearby called adaptors could be tethered and cytoplasmic proteins phosphorylated. This ultimately activates a series of signaling cascades [16,17]. FGFR substrate 2 (FRS2) is one of these adaptors. After FGF binding the FGFRs form dimers and FRS2 adaptor binds to the complex, by which a series of downstream signaling cascades occur leading to the activation of important tumorigenic pathways. Among such activated tumour-leading pathways are the phoshoinositide 3 kinase Protein Kinase A (PI3K-AKT) [18] and the Mitogen-Activated Protein Kinase (MAPK) [19]. In addition, FGFR on its own is connected to phospholipase C-gamma (PLC-γ), in a mechanism that is FRS2 independent and it is also capable of activating Protein Kinase C (PKC) [20]. Notable, PKC can phosphorylate RAF, making the process of MAPK pathway activation to occur more effectively [21].

The downstream molecules of the FGFR pathway, described in more detailed in our previous review [2], constitute actionable targets that are captivating attention for the development of novel antibodies and/or small compounds against cancer-driver mutations in FGFRs and associated signaling molecules, to develop innovative anti-cancer drugs [22] (Figure 1).

### 1.3. The Balancing of FGFR Cascade

In order to have a balanced FGFR cascade, first the FGF signaling should be properly regulated. However, this process is poorly known and may vary based on the cell type [3]. Nevertheless, ubiquitination mediated internalization [14,23], negative regulation through *Spred*, *Sef* and *Spry* genes [15,24,25], are important negative feedback mechanisms balancing the FGFR/FGF axis cascade. Receptor auto-inhibition is another mechanism of control [26,27]. Moreover, heparane sulfate (HS)-binding site and the FGFR’s acid box binding leads to a receptor binding closed conformation, an auto-inhibition mechanism [7,27]. This mechanism blocks the binding of FGF to FGFR. FGFs would therefore bind to other RTKs [28].

## 2. FGFRs as Oncogenic Drivers

FGFRs’ signaling pathways deregulation can work as cancer driving oncogenes, as evidenced by large series of experimental results gathered from experiments conducted with several types of tumors [12,13]. Therby, deregulation of FGFR’s cascade leads to a blockade of apoptosis, an increase of mitogenesis and fosters epithelial-to-mesenchymal transitions [29]. Mechanisms of deregulation are the following ones: (i) expression of fusion proteins with FGFR resulting from gene-translocations that constitutively activate the kinase activity of FGFR [30]; (ii) overregulation of genes and post-transcriptional events, ultimately increasing protein FGFR levels [31]; (iii) high expression levels of FGF outside the cellular matrix, inside the stromal and tumour cells, activating the paracrine/autocrine pathway [32]; (iv) *FGFR* alternative splicing, change of its isoform, processes that ultimately change the ligand-to-receptor specificity, increasing therefore the range of FGFs that are capable of inducing cellular growth [33]; and (v) constitutively activating mutations of *FGFR* (Figure 2). Sarabipour et al. proved that the mutations constitutively activating the receptors give them the capacity to dimerize without ligands triggering them at physiological conditions, as described in (v) [34]. The stabilization of such unbound dimers takes place by a connection between the transmembrane and intracellular domains [35]. It is important to note that the phosphorylation of FGFRs that did not bind is retained, explaining thereby how the high expression level of FGFR leads to cancer formation [10,36,37,38]. However, changes in the structure induced by the binding of the ligand to receptors that had dimerized inside the intracellular region of the membrane can change the transmembrane structure, which will switch its structural conformation into a specific one [39]. Based on transmembrane configuration of dimers, the activity of the receptor is concerted [40]. Quite different are the FGF1 and FGF2 ligands-induced conformational changes. Those ligands modify the distances between the intracellular domains, changing therefore the amount dimers become phosphorylated [41]. For this reason FGFRs are found in different configurational states, even after being bound by the ligand, some of them form an active state while others are intentionally inactivated [34]. Such series of mechanisms inside the cells fine-tune the FGFR signaling cascade of events, leading to apoptosis, mitogenesis, proliferation, epithelial-to-mesenchymal-transitions, and oxygen deliver vessels formation in embryogenesis or tumorigenesis. There are different *FGFR* germline mutations in humans. Mainly they are GOFs in genes that give rise to hereditary diseases, such as osteochondrodysplasias, and tumorigenesis [42]. Fascinatingly, there are mutations all over the gene encoding for FGFR, and are not limited to the kinase region [43]. A peculiar aspect of the signaling pathway involving this receptor is the dependency on the type of tissue [44]; each tumor type has different aberrations in different FGFRs [45,46,47]. Here we focus on breast cancer fibroblast growth factors genes aberrations.

## 3. *FGFR* Genetic Alterations in Breast Cancer

In 1991 Adnane et al. discovered for the first time amplifications of the genes encoding for the fibroblast growth factor receptors in the context of BCs [48]. Since then, several studies confirmed this data proving that the receptors have oncogenic roles and they were further able to explain—to an extent—through which mechanisms these molecules that sit on the cell membranes achieve this purpose. Additionally, point mutations (single nucleotide polymorphisms), high expression of ligands and FGFRs had also identified in previous studies, paving the way for an hypothesis where multiple molecular mechanisms could be leading to an overactivation of the receptors [49]. Deregulation of the fibroblast growth factor molecular cascade in human cancers is frequent. Interestingly, the most common alteration for these receptors is related to the first receptor –FGFR1 (located on the 8p11-12 region of the chromosomal)-while the other five receptors are barely found over-amplified [10,50].

### 3.1. FGFRs Gene Amplification

Around 15% of breast cancers have an *FGFR1* gene mutation [38,43,51]. High expression levels of the *FGFR1* gene and/or of the entire region of the 11q12-14 chromosome (which entangles *FGFR3-4*, and *CCND1*) was observed in 27% of patients positive for Human Epidermal Growth Factor Receptor 2 (HER2), as well as in 23% of patients positive for hormone receptor (HR+) and in 7% of patients who were Triple Negative, (TNBC) meaning they do not express any of the hormone receptors (progesterone and estrogen) nor HER2. The amplifications could be used as prognostic markers for patients, since they were found associated with early relapse and lower survival rates [52,53,54,55,56]. Andrè et al., using cell lines proved that cells with *FGFR1* amplified have to induce overexpression of the gene to support B-cell line’s survival, which is an oncogenic signature. The authors used a couple of commercially available cell lines with *FGFR1* or *FGFR2* amplified, MDA-MB-134 and SUM52, respectively. They proved in both cells that dovitinib (also known as TKI 1258), which is an anti-*FGFR1* antibody, was capable to reduce downstream signalling by a down modulation of pFRS2 and pERK/MAPK as shown by western blotting, and in vivo it reduced tumours in primary breast cancer xenografted HBCx-2 mice [57]. The half maximal inhibitory concentration (IC_50_) measures the potency of a pharmaceutical construct by an equation standing for the amount of a specific drug required to inhibit a chemical or biological function in vitro. To attain such value for TKI 1258 in the two cell lines, SUM52 and MDA-MB-134, 180 nmol/L and 190 nmol/L were needed, respectively. As expected, in control cells that did not express none of the two receptors (*FGFR1* nor *FGFR2*) the IC_50_ values were higher and above 2000 nmol/L. The authors showed that the administred drug (50 mg/kg) to the HBCx-3 mice brought to a tumour regression vs. empty control (*p* < 0.001) [57]. Moreover resistance to hormone therapies can be driven by *FGFR1* amplifications. As a matter of fact, Turner et al., demonstrated in MDA-MB-134 and SUM44 BC cell lines, overexpressing *FGFR1*, a resistance to the endocrine treatment, namely the 4-hydroxytamoxifen (4-OHT). The authors used cell TiterGlo assay to prove survival of the cells transfected with or without small interfering RNA targeting FGFR1 (*siFGFR1*), to silence the gene [38]. Treatment with *siFGFR1* reversed such resistance. As a conclusion the authors evinced that FGFR1 is an important factor driving sensitivity to 4-OHT therapy. Another experiment adding to such proof-of-concept—elucidating that among the FGFRs specifically *FGFR1* is responsible for the sensitivity to the drug—is the one where it was shown that the addition of FGFR2 to si*FGFR1*-treated cell lines was not capable to achieve the same results. The cells in the latter scenario were still resistant to 4-OHT. Therefore the resistance was FGFR1-dependent. Furthermore, FGFR1 inhibitor (PD173074) triggered a loss of colony formation ability of the cell lines, suggesting that *FGFR1* confers transformative capabilities to the cells. This is in line with the evidences proving the oncogenic role of FGFR1 in breast cancer [38].

In the MONALEESA-2 clinical trial, based on the combination of letrozole with ribociclib (a Cyclin Dependent Kinase 4/6, CDK4/6, inhibitor), *FGFR1* amplification was related to a lower PFS respect to wild type patients for the *FGFR1 gene*. In in vitro studies focused on estrogen receptor positive BC patients with amplification of *FGFR1*, it has been shown that there is a correlation between *FGFR1* amplification and resistance to CDK4/6 inhibitors ribociclib or palbociclib. Such resistance-to-treatment was overcome by the combination of anti-FGFR drugs, such as lucitanib. Moreover the combination of CDK4/6 inhibitor palbociclib and FDA-approved FGFR inhibitor erdafitinib showed a complete in vivo response in SCID/beige mice xenografted with estrogen receptor positive and *FGFR1* amplified breast cancers. Moreover they showed that circulating tumor DNA from 34 patients progressing to the CDK4/6 drugs had amplifications or activating mutations of FGFR1 and FGFR2, which were as high as 41% in the patients [58]. Hence, combining treatments acting on both pathways, namely the FGFR/FGF and the CDK4/6 pathways, could prove a great option to overcome resistance to CDK4/6 inhibitors.

Different investigations have proven that amplifications of the *FGFR*2 (occurring in less than 5% of triple negative breast cancers), and other mutations activating the FGFRs have been related with maintenance of tumor-initiating cells and a high sensitivity to FGFR inhibitors [59].

### 3.2. FGFRs Activating Mutations

Besides the oncogenic role that amplifications of the *FGFR* gene have, a least common form of alterations causing FGFR-driven BC are mutations that constitutively activate the receptor [10]. There are different mechanisms by which *FGFR* activating mutations could lead to aberrant FGFR signaling. These ones include the following ones: (i) dimerization of FGFRs that become bound in a form that will not be reversible; (ii) receptor kinase domain over activation; and (iii) alterations of binding affinity between FGFR to FGF. In fact, the most frequently occurring oncogenic FGFR aberrations in BC are *FGFR*1 translocation [60] and *FGFR*1 amplification leading to activating mutations (10–15%) [38,52,53,54,55,56]. These two genetic aberrations can modify the phenotype of BC cells transforming them onto a phenotype exhibiting more sensitivity to FGFR inhibitors [55]. They can also drive the endocrine resistance, as explained in the previous chapter [38]. The translocation of the *FGFR*2 gene leading to activating phenotypes, can also bring to cellular transformation and make the cells more anti-FGFR sensitive [60]. In addition, *FGFR2* amplification, present in less than 5% of BC cases, can make cancer cells more resistant to FGFR inhibitors, as seen in pre-clinical models [57,58,59]. Many interesting activation mutations have been identified in solid tumors, such as BC, and among them:
In FGFR1: *in vitro* a couple of point mutations (K656E and N546) can affect FGFRs intracellular domain, constitutively activating the receptor [61,62].In FGFR2: there are 12 mutations reported in a Catalog of Somatic Mutations in Cancer (COSMIC), which is the largest database entailing breast cancer somatic mutations. There are only seven missense mutations capable of constitutively activate the receptor. Among these, the most common ones in *FGFR2* are N549K, S253R and P253R [43]. Moreover these three activating mutations are located on the extracellular region of the receptors between the two immunoglobulin-like domains, domains that are important for ligand binding [63]. In estrogen receptor positive breast cancers the M538I and N550K mutations of *FGFR2* contribute to giving resistance to inhibitors of SERDs and CDK4/6. Moreover, in some cohorts of estrogen receptor positive MBC patients resistant to CDK4/6 and SERDs *FGFR2* mutations were detected. This could imply that FGFR2 could be involved in a mechanism conferring some resistance to patients. Therefore *FGFR2* mutated patients could benefit most from the combination of CDK4/6, SERDs and FGFR inhibitors.FGFR3: from the COSMIC database, 13 point-mutations were detected. Among them, S249C, R248C, G370C, K650E, R399C and Y373C were the most frequent ones. Frequent activating mutations in this gene affect either the extracellular (R248C, S249C) or the transmembrane (G370C, S371C, Y373C, G380R A391E) protein domains. There are also a number of rare mutations within the kinase domain, such as, K650E, K650N, K650M, K650T K650Q, and N540S [43,64].FGFR4: there are four activating mutations for this gene, which are located within the kinase domain, with a couple of them (K535 and E550) causing auto-phosphorylation of the receptor, and therefore constitutively activating it [43,65].

It is worth mentioning, some of such mutations could be predictive biomarkers for early detection of BC development.

### 3.3. Fusion of FGFRs Genes

A fusion of genes consists in the formation of a hybrid gene by the joining of two different genes through either a chromosomal inversion or translocation. Looking at the whole range of *FGFR* aberrations, gene fusion consists of only 8% of such aberrations [10,43]. In *FGFR1* there are 11 genes in total that could potentially participate in the fusion. Examples of these fusion genes are *FOP, BCR*, and *ZNF198*. The most common fibroblast growth factor receptors implicated in gene fusions are *FGFR2* and *FGFR3*. The most notorious gene fusions are those observed in myeloproliferative syndrome patients. The gene fusion of FGFR3 with TACC3 (FGFR3-TACC3), constitutively activates the receptor [66,67]. In BC, *FGFR1-3* fuses with many gene partners (i.e. *AFF3, AHCYL1, BAIAP2L1SLC45A3, BICC1, PPAPDC1A, TACC1, TACC2, TACC3, NPM1*) [43,68,69].

### 3.4. Genome-Wide Studies

Genome-Wide-Association-Studies (GWAS) has brought evidence about the potential of risk prediction for the development of BC in individuals with Single Nucleotide Polymorphisms (SNPs) in the second intron of *FGFR2* gene [70,71,72,73]. Easton et al., for example, in a cohort of 4,398 breast cancers vs. 4,316 healthy individuals used GWAS to investigate common SNPs to find risk-associated factors [70]. Five new loci were found to be significantly associated with BC (*p* value < 10^−7^). *FGFR2* was one of these loci, thereby corroborating the potential oncogenic role of alterations in this gene in BC [70]. According to well-powered GWAS, conducted by Stacey et al., rs4415084 and rs1094179 SNPs (located on chromosome 5p12) were associated with increased risk of encountering breast cancer. The difference reached the highest statistical significance ER + BC (*p* value = 1.3 × 10^−17^) [71]. Meyer et al. proved with microarray gene expression analysis that *FGFR2* is expressed at higher levels in rare homozygotes [72]. The authors then confirmed the data by Real-Time PCR (RT-PCR). They showed that patients with the rare homozygous had higher levels of *FGFR2* as compared to that of normal homozygotes (Wilcoxon *p*- value = 0.028). They also showed that this difference was attributable to a change in the promoter-binding site for Oct-1/Runx2 [72]. Easton et al. [70], Hunter et al. [73] and Stacey et al. [71] demonstrated an association between alleles in *FGFR2* with a higher chance of developing sporadic post-menopausal BC. Notably, the Hunter et al. [73] study made of 1,145 postmenopausal European women and 1,142 healthy controls investigated 582,173 SNPs [73]. In their experiment they showed that alterations in different locations of the genome correlated with the malignancy. The most significant ones were rs2420946, r1219648, rs2981579 and rs11200014. All of them were on the chromosome 10q26.13 and on intron 2 of *FGFR2* position.

Kim et al. provided a plausible explanation. They showed that the *FGFR2* gene fosters BC by maintaining a population of cells that have the capacity of initiating tumours, namely Cancer Stem Cells (CSCs) or tumor-initiating cells (TICs) [59]. In fact, the authors demonstrated in BC, that CD29^high^ CD24^+^ TICs expressed significantly higher levels of *GABRA4*, *FGFR2* and *FOXA1* mRNA expression. Additionally, down-regulation of *FGFR2* by short hairpin RNA (shRNA), which is a short molecule of RNA engineered to silence target genes (in this case *FGFR2*) via RNA interference, in mouse models substantially reduced (64–70%) the CSCs subpopulation CD29^high^ CD24^+^. Interestingly, the non-TIC cells (CD29^low^ CD24^−^) subpopulation was significantly increased (65–67%) after using sh*FGFR.2*. Therefore a down modulation of FGFR2 could cause a non-TICs increase and a TICs decrease [59]. Furthermore, they proved that in mice treated with sh*FGFR2* there is an increased of bipotent precursor-like cell population (K18+K14+). The generation of bipotent populations generated by *FGFR2* knockdown could be overcome by FGFR2. Therefore, a valid strategy may consist in the inhibition FGFR2 in order to decrease those BC CSCs. Kim et al. additionally proved that tumour growth could be inhibited by treatment with FGFR2-inhibitor (TKI258) using NOD/SCID mouse models xenografted with breast cancer tumours overexpressing FGFR2. Such inhibition of tumour growth was followed by a significant reduction of protein FGFR2 phosphorylation together with Erk1/2 activation. This further proved that the inhibition was FGFR2-dependent [59]. Guagnano et al., through a screening that included BC cell lines with FGFR alterations, studied cell sensitivity to NVP-BGJ398, which is an anti-FGFR inhibitor. They focused on nine types of well known *FGFR* genetic alterations from literature: *FGFR*1-*FGFR*2 chromosomal translocations; *FGFR1-3* activating mutations; *FGFR1-4* copy number gains. The drug was evinced as a powerful multi-kinase inhibitory molecule against Vascular Endothelial Growth Factor Receptor (VEGFR) 2, besides FGFR1-4. Finally the experiments of these researchers demonstrated a predictive role for such alterations in the FGFR genes for response to NVP-BGJ398 therapy [60]. In a small study of 13 lobular BC, Reis-Filho et al. proved the expression of high copy number levels on the 8p12-p11.2 chromosomal locations, in six patients (46% of cases) [55]. Moreover, using small-interfering molecules against *FGFR1* in SU5402 cell line, the authors demonstrated that FGFR1 inhibitor could block breast cancer survival of ductal breast adenocarcinoma cell line MDA-MB-134 [55]. In summary different research analyses of various research groups support that using drugs blocking the FGFR/FGF signaling pathway is a good approach that is worth experimenting at later stages of clinical development involving randomized-to-control patients. Accordingly to Next Generation Sequencing (NGS) investigating FGFR levels in breast cancer, low levels of *FGFR3* and *FGFR4* were detected. As a matter of fact, the NGS study of Helsten et al. investigated their expression levels in 4853 solid tumors, with 522 breast cancer. In this experiment they showed a very low level of amplification of FGFR3 and *FGFR4,* lower than 1% and 2%, respectively [43]. Conversely, in a RT-PCR investigation of 103 breast-tumor samples and 10 tumor cell lines *FGFR3* was not detected at all and *FGFR4* was present in a maximum of 32% of the total BC population [74].

## 4. Anti-FGFR Therapies

There is a growing interest in FGFR/FGF inhibitors to block the formation and progression of BC in developing new targeted therapies against this pathway [75]. Clinical evaluations have been conducted over small FGFR inhibitors, selective or nonselective, even though many are early clinical trials [57]. Novel drug development should be focused in the attainment of an increased selectivity to the FGFRs ATP-binding domain located in the intracellular region in order to reduce to the maximum extent the toxicity [45]. BGJ398 (infigratinib) is a pan-FGFR inhibitory molecule that has been evaluated on its own (NCT01004224)—to establish its maximum tolerated dose (MTD) for primary outcome and ORR together with pharmacokinetics and pharmacodynamics for secondary outcome measures [76], respectively—was recently completed. In the 67 enrolled patients, ORR was 25.4% and DCR 64.2% [77]. Additionally, MTD for BGJ398 with BYL719 was investigated in another clinical investigation on phase I (NCT01928459) *FGFR 1–3* and *PIK3CA* mutations bearing solid cancers, which was recently completed and whose results have not been posted yet. AZD4547 is an additional tyrosine kinase inhibitor, whose activity had been previously shown to be strong for FGFR-3. On the other its activity against *FGFR4* was very low. A phase I is currently investigating safety and efficacy of this compound in endocrine progressing BC patients bearing polisomy or amplification of the *FGFR1* gene (NCT01791985), which was recently completed and whose results are eagerly awaited. Another phase I study (NCT03238196) has been evaluating in ER+ HER2- MBC patients the FGFR inhibitor, called erdafitinib, together with palbociclib and hormone therapy fulvestrant.

A phase 2 clinical trial (NCT04125693) investigates the oral pan-FGFR inhibitor rogatanib as second line medication of solid tumours, including BC. A phase 1 and 2 dose expansion clinical trial (NCT02052778) investigates the oral selective irreversible pan-FGFR inhibitor futibatinib as second line treatment of advanced solid tumours, including metastatic breast cancer. A phase 2 clinical trial (NCT0402446) investigates the oral pan-FGFR inhibitor rogatanib as second line treatment as monotherapy or together with fulvestrant.

In preclinical studies multi-kinase inhibitors, capable of targeting also FGFRs together with other tyrosine kinases, have been showing promising results [7]. A phase I trial investigating such inhibitors has shown great positives as to safety and tolerability of this drug type. Dovitinib (TKI258) showed its effectiveness in targeting FGFR1-3, PDGFR and VEGFR1-3 [78], and it has been experimentally used for the treatment of HER2-negative MBC in combination with fulvestrant. However this clinical trial was too slow and had to be therefore terminated (NCT01528345).

E3810 (lucitanib), against colony stimulating factor 1 receptor (CSF1R)—3, FGFR1, FGFR2 and VEGFR1 have been studied on their own in two phase II clinical trials involving MBC patients with the presence or absence of *FGFR*1 amplification; one of them was a phase 2 study (NCT02202746), which was terminated by the sponsor of this clinical trial. Recently, the safety and tolerability of a triple kinase inhibitor (FGFR, PDGFR and VEGFR) was investigated in a phase 1 study made of 19 estrogen receptor positive MBC postmenopausal women (NCT02619162). At PR2D (nintedanib with letrozole) there was a 55% mean increase in the plasma levels of FGF23 and there were no detectable levels of 17-B-estradio in the plasma of patients [77].

Other undergoing strategies inhibiting FGFR/FGF inhibitors together other signaling pathways exist [79], but these were not the main focus of this review. For a reference to such combinatorial strategies refer to our previous publication [2]. Antibodies against FGFR isoforms represent a valid therapeutic strategy to intervene in BC. As a matter of fact, GP369 recognizes FGFR-IIIb isoform and has exhibited good results in blocking breast cancer cell line proliferation [80]. Such positive preliminary results warrant further research. Lastly, another approach against the FGFR/FGF axis concerns the use of inhibitors of FGF ligands. Long pentraxin-3 (PTX3) is an inhibitor of various FGFR ligands, among them FGF2 and FGF8b, which have both been found to be implicated in breast cancer development [81]. FP-1039 is a recently developed ligand-trap in which a ligand-binding domain of FGFR1 is fused to an Ig-Fc domain. This compound showed promising activity in vitro and passed a phase I clinical trial (NCT00687505) for solid tumors, including breast cancer [82]. Table 1 outlines all the ongoing clinical investigations of anti-FGFRs therapies. Although such results are crucial for the anti-FGFR therapy development, more knowledge of the molecular mechanisms by which FGFR function and lead to breast cancer in correlation with other well-known molecular pathways are also eagerly awaited and important to best design new treatments and to best give the most effective ones to each patient.

## 5. Discussion and Conclusions

The established BC oncogenic driver FGFR has been found involved in various tumor-related roles, leading to angiogenesis, tumor growth and apoptosis avoidance. There are different FGFR variations correlating with breast cancer. Therefore many different strategies have been designed in order to block the FGFR/FGF axis. For this purpose phase 1/2 clinical investigations have been investigating several therapies targeting the FGF/FGFR axis. Among such molecules, some examples are futibatinib (TAS-120), nindetanib, rogaratinib (BAY-1163877), erdafitinib, an anti- pan-FGFR molecule infigratinib (BGJ-398), dovitinib and FGFR1–3 inhibitor AZD4547. Notably, *FGFR1* genomic aberrations are the most common ones, while gene amplifications in the FGFR2-6 and mutations constitutively activating the FGFR are uncommon. Thus, among the different targets for future therapies against FGFRs, FGFR1 should the considered as the primary one to be further pursued. A combination of anti-FGFR therapies should be experimented with other drugs targeting downstream pathways of the FGFRs/FGFs axis, mutation-bearing antigens and other Tyrosine Kinase cell membrane receptors such as AXL, CCK, EGF, HGF, PDGF, LMR, RET, TIE, RYK, ROS, and VEGF. With the impelling advancements of personalized medicine in oncology, stratification of patients - based on definite molecular modifications - and the discovery of always more precise biomarkers predictive both disease occurrence and treatment efficacy are leading to the development of accurate molecular-based methods aiding clinicians to choose the right therapy, or combination of therapies, for each individual patient. Furthermore, as immunotherapy is today standing at the front-line of innovation of anti-cancer treatments, it would be curious to test anti-FGFR or anti-ligand FGF drugs together with immunotherapeutic agents—like PD-1/PD-L1 or CTLA4-CD20 checkpoint inhibitors—to improve both survival and quality of life of breast cancer patients through new and more precise strategy of fighting cancer, focusing on checkpoints or drugs targeting driver mutations presented by cancer cells on their surfaces and beyond.

In conclusion, *FGFR* alterations occur in about one of seven breast cancer patients, which represent a large portion of cancer patients, since the disease is very common and unfortunately still represents a major killer. The knowledge that has been gained on the structure of the receptor and its signaling pathway has always led to the development of better drugs against it. The hindsight from clinical trials recently conducted is pushing towards the direction of using combination of therapies to overcome a single drug resistance. Most interestingly the results from MONALESA-2, showing that patients with lower PFS to CDK4/6 inhibitor presented amplifications of *FGFR1*. Therefore, FGFR1 could be conferring the cancer cells the capacity to proliferate and a good strategy could be that of inhibiting both FGFR/FGF and the CDK4/6 pathways. More clinical trials testing FGFR inhibitors in combination with other drugs are warranted.

## Figures and Tables

**Figure 1 ijms-21-02011-f001:**
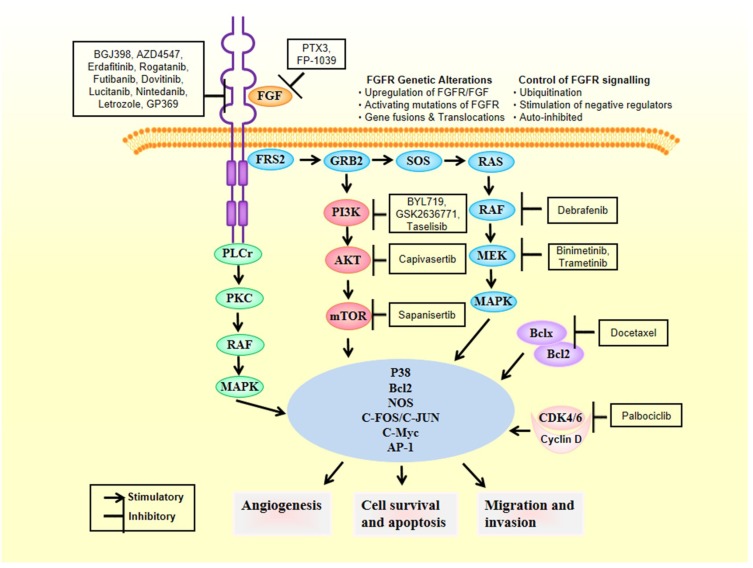
Therapies fighting breast cancer through the fibroblast growth factor receptor (FGFR) pathway.

**Figure 2 ijms-21-02011-f002:**
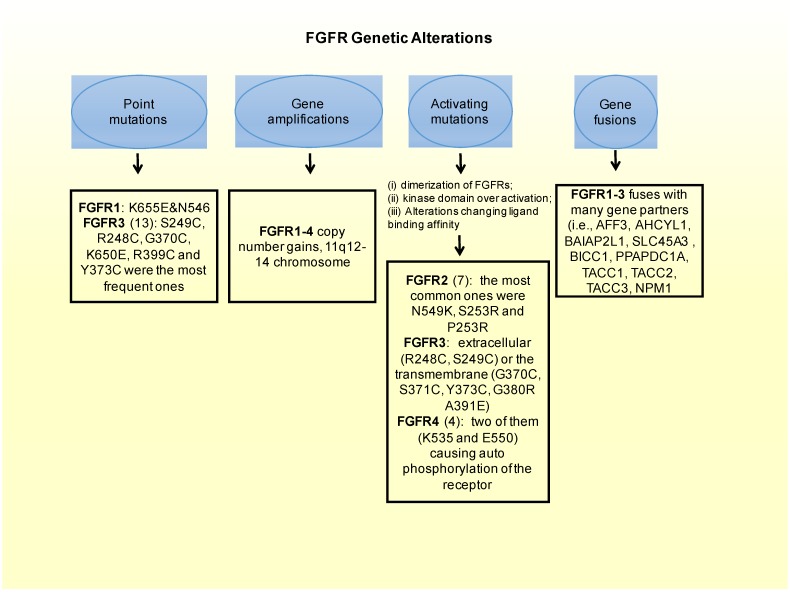
FGFR genetic alterations leading to breast cancer.

**Table 1 ijms-21-02011-t001:** Ongoing clinical trials investigating anti-FGFRs in the breast cancer pathology.

Clinical Trial Identifier Code	Investigation Plan	Drug/s	Clinical Setting Line	Primary Endpoint	Stage of Development	Clinical Trials Status
NCT04125693	50 participants, Single Group Assignment, Open label	Rogaratinib (800 mg twice daily)	Second line	TEAEs	2	Enrolling by invitation
NCT02052778	371 participants, Single Group Assignment, Open label	Futibatinib (dose escalation)	Second line	ORR and EPR	1 and 2	Recruiting
NCT04024436	168 participants, Non-Randomized, Open label	Two arms design: Arm 1: Futibatinib (orally given every 28 days); Arm 2: Futibatinib (orally given every 28 days) plus Fulvestrant (intramuscularly given every 28 days)	Second line	ORR, CBR and PFS	2	Active, not recruiting
NCT03238196	32 Participants, Non-Randomized, Open label	Fulvestrant plus Palbociclib plus Erdafitinib in a dose-escalation design (Fulvestrant 500 mg once daily plus Palbociclib 125 mg once every 21 days followed by 1 week of rest and Erdafitinib 4 to 8 mg once daily).	Second line	Safety and Tolerability	1	Recruiting
NCT02465060	6452 participants, Non-Randomized, Parallel assignment, Open label	Adavosertib, Afatinib, Binimetinib, Capivasertib, Crizotinib, Dabrafenib, Dasatinib, Defactinib, AZD4547, Larotrectinib, Nivolumab, Osimertinib, Palbociclib, Pertuzumab, GSK2636771, Sapanisertib, Sunitinib malate, Taselisib, Trametinib, Trastuzumab, Trastuzumab emtansine, Vismodegib	Second line	OR	2	Recruiting
NCT03344536	55 participants, Single group assignment, Open label	Fulvestrant (initially administered 500 mg at intervals of 1, 15, 28 days and then after 3 days) and Debio 1347 (administered every day in a dose escalation manner).	For the phase II they could be first or second line; for the phase I, they be treated multiple times	DLT	1 and 2	Recruiting
NCT02393248	280 participants, Single group assignment, Open label	Combination therapy: Gemcitabine plus Cisplatin plus INCB054828 or Pembrolizumab plus INCB054828 or Docetaxel plus INCB054828 or Trastuzumab plus INCB054828.	Second line	MTD	1 and 2	Recruiting

Abbreviations: Progression Free Survival, PFS; Objective Response, OR; Dose Limiting Toxicity, DLT; Maximum Tolerated Dose, MTD. The information was extracted from clinicaltrials.gov.

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
