# Peer review of "The Fibroblast Growth Factor Receptors in Breast Cancer: from Oncogenesis to Better Treatments"

_ijms, 2020, doi:10.3390/ijms21062011_

Round 1

Reviewer 1 Report

The authors present a comprehensive review about targeting FGFR in breast cancer. In general, is a well-structured with concise information paper.

The manuscript requires a deep English editing. In some areas it´s really complicated to follow it.

Other comments are described below:

  • "As to the epidemiology of the disease, Breast Cancer (BC) is a devastating cancer for females. In the planet it is the first one of all types of tumors in terms of frequency and second in terms of mortality after lung cancer. Only in the United States 271270 patients were diagnosed with BC and the people who died in the continent were 41488 in the year 20192". This paragraph is difficult to understand and give very sharping information. I suggest re-write in a better way.
  • "The FGFR1 gene is located on chromosome 18p11.23, while -2 is in chromosome 10q26.13, -3 is in chromosome 4p16.3, -4 is in chromosome 5q35.2, -5 is in chromosome 4p16.3, and finally -6 is located in chromosome 6p21.33 (also called Fibroblast Growth factor like-1)." Please correct it, -3 is a bit confusing. Is preferable (FGFR3, FGFR4...)
  • "The structure of the receptor has been fully described in our previously published review". It´s preferable to make a mini briefing and not to use this phrase repeated twice in the text.
  • "Mechanisms of deregulation are the following ones: (i) expression of fusion proteins with FGFR resulting from gene-translocations that constitutively activate the kinase activity of FGFR 30; (ii) overregulation of genes and post-transcriptional events, ultimately increasing protein FGFR levels 31...". In my own opinion the authors could explain this mechanism better in a figure form.
  • There is no conclusion.
  • Discussion end suddenly. The authors should rewrite to end it in a better way.

Author Response

Dear Reviewer 1,

Thank you for your suggestion.

Please find the following point-to-point answers:

The authors present a comprehensive review about targeting FGFR in breast cancer. In general, is a well-structured with concise information paper.

Thank you.

The manuscript requires a deep English editing. In some areas it´s really complicated to follow it.

Sobhani N., who graduated in the UK, has just double read it and double checked the language.

Other comments are described below:

  • "As to the epidemiology of the disease, Breast Cancer (BC) is a devastating cancer for females. In the planet it is the first one of all types of tumors in terms of frequency and second in terms of mortality after lung cancer. Only in the United States 271270 patients were diagnosed with BC and the people who died in the continent were 41488 in the year 20192". This paragraph is difficult to understand and give very sharping information. I suggest re-write in a better way.

Thank you for the suggestion. We have re-written this paragraph.

  • "The FGFR1 gene is located on chromosome 18p11.23, while -2 is in chromosome 10q26.13, -3 is in chromosome 4p16.3, -4 is in chromosome 5q35.2, -5 is in chromosome 4p16.3, and finally -6 is located in chromosome 6p21.33 (also called Fibroblast Growth factor like-1)." Please correct it, -3 is a bit confusing. Is preferable (FGFR3, FGFR4...).

Thank you for the suggestion. We have fixed it.

  • "The structure of the receptor has been fully described in our previously published review". It´s preferable to make a mini briefing and not to use this phrase repeated twice in the text.

Thank you for the advice. We have added a brief summary of the structure of the receptor.

  • "Mechanisms of deregulation are the following ones: (i) expression of fusion proteins with FGFR resulting from gene-translocations that constitutively activate the kinase activity of FGFR 30; (ii) overregulation of genes and post-transcriptional events, ultimately increasing protein FGFR levels 31...". In my own opinion the authors could explain this mechanism better in a figure form.

We have added a Figure 2.

  • There is no conclusion.

We have just added a conclusion paragraph in the discussion. We renamed the last part as “Discussion and Conclusions”.

  • Discussion end suddenly. The authors should rewrite to end it in a better way.

We have just written a new conclusion at the end of this section to briefly summarise the content.

Reviewer 2 Report

The manuscript of Navid and coworkers was aimed at reviewing the involvement of FGF receptors in Breast Cancer and their potential role as useful target in the disease therapy. Although of interest, content is not original as most of the information has been already provided in a similar paper published in 2018 (Cells 2018, 7, 76; doi:10.3390/cells7070076).  In my opinion the manuscript should be rewritten focusing on a critical appraisal of the current therapy taking into account the clinical aspects (e.g. stratification of patients) and how a targeted therapy could ameliorate patient’s quality of life and survival. The title of the manuscript is somewhat misleading as the pharmacology of monoclonal antibodies is only marginally described

Author Response

Dear Editor,

Thank you for your comments. Please find the following point-to-point answers to your suggestions:

The manuscript of Navid and coworkers was aimed at reviewing the involvement of FGF receptors in Breast Cancer and their potential role as useful target in the disease therapy.

Thank you. That was exactly the purpose of this invited manuscript.

Although of interest, content is not original as most of the information has been already provided in a similar paper published in 2018 (Cells 2018, 7, 76; doi:10.3390/cells7070076).  

We have cited it. We have re-written our paper creating as much as possible a new version, with a new equip of colleagues in an outstanding academic institution such as Baylor College of Medicine where we research. Since some parts of the old paper were lacking some in-depth information we decided to write this new and improved what was old. Moreover this article is a review of the literature coming from the latest advancements in the field of FGFR in BC over the past 2 years and the ongoing clinical trials. We believe it is important for the field reviewing the latest therapies and ongoing clinical trials, while commenting on the ones that have been completed and giving an updated medical and biological opinion based on the recent data from the literature since 2018.

In my opinion the manuscript should be rewritten focusing on a critical appraisal of the current therapy taking into account the clinical aspects (e.g. stratification of patients) and how a targeted therapy could ameliorate patient’s quality of life and survival.

We have rewritten the invited manuscript with this purpose, focusing on the latest therapies for FGFR to improve patients’ quality of life and survival (OS, PFS, ORR) based on the current knowledge from clinical trials up to February 2020. The table summarizes what has been reported on clinicaltrials.gov and we are glad to discuss its content in the full chapter 4 “Anti-FGFR Therapies” as you mentioned. Moreover, to facilitate the reader and in accordance with what suggested by Reviewer 1, we have added a Table 2 to explain FGFR genetic alterations driving breast cancer formation.

The title of the manuscript is somewhat misleading as the pharmacology of monoclonal antibodies is only marginally described.

We have changed the title to better suit the article’s theme as you suggested.

Round 2

Reviewer 1 Report

The authors have improved substantially the manuscript with the changes suggested.
I have no additional comments

Reviewer 2 Report

The manuscript has been improved therefore it is now suitable for publication